# Design and Economic Assessment of Alternative Evaporation Processes for Poly-Lactic Acid Production

**DOI:** 10.3390/polym14102120

**Published:** 2022-05-23

**Authors:** Jonghun Lim, Hyungtae Cho, Kwon-chan Son, Yup Yoo, Junghwan Kim

**Affiliations:** 1Green Materials and Processes R&D Group, Korea Institute of Industrial Technology, 55, Jonga-ro, Ulsan 44413, Korea; ljh94@kitech.re.kr (J.L.); htcho@kitech.re.kr (H.C.); yooyup@kitech.re.kr (Y.Y.); 2Department of Chemical and Biomolecular Engineering, Yonsei University, 50, Yensei-ro, Seoul 03722, Korea; 3Advanced Materials Production Department, KCC Corporation, 19, Wanjusandan 4(sa)-ro, Jeonju 55321, Korea; kcson@kccword.co.kr

**Keywords:** poly-lactic acid, multiple-effect evaporation, thermal vapor recompression

## Abstract

In this work, alternative evaporation processes for PLA production were designed with economic assessment. The suggested processes are the multiple-effect evaporation (MEE) process and thermal vapor recompression (TVR)-assisted evaporation process. First, the MEE process can efficiently reuse waste heat by additional column installation, thereby reducing the steam energy consumption. The proposed MEE process involves five columns, and after the evaporation in each column, the waste heat of the emitted vapor is reused to heat steam in the reboiler of the next column. Second, the suggested TVR-assisted evaporation process utilizes an additional steam ejector and recovers waste heat from the emitted vapor by increasing the pressure using high-pressure driving steam at the steam ejector. Each alternative process was modeled to predict the steam energy consumption, and to determine the cost-optimal process; the total annualized cost (TAC) of each alternative process was calculated as evaluation criteria. In the simulation results, the alternative processes using MEE and TVR reduced the steam consumption by 71.36% and 89.97%, respectively, compared to the conventional process. As a result of economic assessment, the cost-optimal process is the alternative process using TVR and the TAC can be decreased by approximately 90%.

## 1. Introduction

Among the products produced using lactic acid (LA), poly-lactic acid (PLA) is a bioplastic that could replace petrochemical polymers [1,2]. PLA is also considered an eco-friendly plastic as it is biodegradable and compostable, as it is extracted from renewable resources, such as sugar and starch [3,4]. In general, the PLA production process consists of a prepolymer section, a lactide section, and ring opening polymerization (ROP) [5]. In the pre-polymer step, crude LA is placed into the evaporator to remove water, and after evaporating the water, the concentrated LA is introduced into the pre-polymer reactor for reaction [6,7,8]. After the reaction has been completed, in the lactide step, the pre-polymer is placed into the lactide reactor for polymerization [9,10,11]. In particular, the evaporation process of the pre-polymer section, which is a process of concentrating low-purity LA into high-purity LA, is considered a high-cost process, as a large amount of steam energy is consumed in the evaporation of low-purity LA [12]. In addition, in many multiple-unit operations, utilities, such as steam and electricity, are used across multiple-unit operations and recycled from one unit operation to another, but in PLA production, steam and electricity are not generated by other processes and, thus, the cost of utilities is significantly high. Thus, the use of PLA is currently hindered by low economic efficiency due to the low productivity and high cost of the production process.

To solve this problem, studies are being actively conducted to improve the productivity of PLA and reduce the cost of the production process. Park et al. attempted to reduce energy consumption by designing a PLA production process by changing the separation method using the SSO-88 catalyst for the lactide section [5]. The lactide yield was maintained at 94% in the rapid reaction state, and energy consumption was reduced. Tong et al. increased the yield and productivity of LA through purification using an anion exchanger, amberlite IRA-92, of feed using paper sludge as a cellulose feedstock [13]. A yield of 82.6% was achieved by increasing the PH to the range of 5.0 to 6.0 using the anion exchanger. Although LA purification using solvent extraction is simple and enables continuous operation, there are problems associated with the use of chemical substances and the drop in distribution coefficients. Madzingaidzo et al. focused on the fact that amino acid migration is similar to LA and concentrated the LA to 150 g/L through purification using mono- and bi-polar electro dialysis [14]. There were, however, problems with membrane fouling and polarization, and a large amount of electricity consumption and inefficiency in the production process were also major disadvantages. Oscar et al. obtained LA with significantly improved purity (80.1%) compared to that achievable in the existing process by LA purification using a reactive distillation column [15]. However, there were problems of high installation cost and high energy consumption.

Despite the PLA productivity improvements and PLA production cost decreases realized thus far, the existing research has not yet overcome problems, such as cost increases, due to the addition of chemicals, process stability deterioration, high installation cost, and high sulfuric acid consumption. Due to their inability to allow continuous operation, there is a limit to their applicability for the PLA production process.

To address this problem, alternative evaporation processes for PLA production were designed in this research, and economic assessment was performed. The suggested alternative processes are the multiple-effect evaporation (MEE) process and thermal vapor recompression (TVR)-assisted evaporation process. Each alternative process was modeled to predict steam energy consumption. Then, the cost-optimal process was determined through techno economic analysis by calculating the total annual cost (TAC). The objectives of this work were to identify a cost-optimal alternative evaporation process for PLA production and to increase PLA usage in many industries for environmental protection by reducing the use of plastic, which is fabricated from petrochemicals. The novelty of our work can be summarized as follows. (1) This research represents the first attempt to decrease the cost of the evaporation process in PLA production by using MEE and TVR to recover waste energy efficiently from emitted vapor. (2) The results will enable PLA usage to be increased in many industries by providing a cost-optimal alternative evaporation process for PLA production. Hence, this work provides alternatives for handling environmental problems caused by petroleum-based plastic. (3) Finally, this work provides valuable insight into the many commercial industries of PLA production for achieving cost effectiveness and environmental protection.

## 2. Methodology

This section describes the methodology used to design the alternative processes for evaporation in PLA production. The suggested alternative processes are the MEE process and TVR-assisted evaporation process, and the alternative and conventional process models were developed by using Aspen Plus V10.0. Further, the specifications of each model are referred to as operating conditions of actual commercial plants.

### 2.1. Thermal Dynamic Model and Assumptions

The UNIQUAC activity coefficient model was used for the thermodynamic equations for process model development, and Haydon O’Connell, among the UNIQUAC models, was utilized to apply the vapor–liquid equilibrium binary parameters of LA and water to the simulation [16,17]. This thermodynamic model is mainly employed to determine the parameters of VLE and LLE in non-ideal chemical systems. In particular, the Hayden O’Connell equation is mainly used in processes involving organic acids because strong association and solvent effects can be considered [18,19].

The assumptions made to develop the models of the existing and alternative processes were as follows.

(1) Ignoring impurities, the composition of the feed was assumed to be LA 10 wt% and water 90 wt%. As it is a prior process for PLA production, the product composition was assumed to be LA 75 wt% and water 25 wt%. (2) In the three processes investigated, the feed was compressed through a pump, underwent heat exchange with steam or water for pre-heating, and was then fed into a distillation column. (3) In all processes, the number of stages of the distillation column was fixed at 11 stages, the feed location was at stage 5, and the reflux stream at the top of the tower entered at stage 1. (4) The steam used as a heat source was middle-pressure steam, and the temperature and pressure were assumed to be 180 °C and 9 kg/cm^2^, respectively. (5) When modeling the distillation column in Aspen plus V10.0, the calorific value of the reboiler could be simulated, but the amount, temperature, and pressure of steam flowing into the reboiler could not. Therefore, in this study, a virtual reboiler was added to predict the steam usage of the modeled distillation column. By setting the amount of heat calculated in the virtual reboiler and the amount of heat used in the distillation column to be the same, the amount of steam used in the distillation column was predicted.

### 2.2. Conventional Process

Figure 1 shows a diagram of conventional process model using single-effect evaporation (SEE). In the SEE process, the feed (FEED) of 90 wt% of water is compressed through the pump (P1) and then pre-heated into a series of heat exchangers (HX-F1, HX-F2). Pre-heating is performed using the waste heat of the vapor discharged from the distillation column during the evaporation process, and the waste heat is recovered by separating it into gas and liquid through a flash drum (FD). Finally, the pre-heated feed goes through an evaporation process in a distillation column (DC), and LA, which has a relatively high boiling point, comes out of the tower together with unevaporated water and has a lower boiling point than LA. Water is discharged in the form of gas to the top of the tower. Through a series of processes, the feed is concentrated to 75 wt% of water and discharged as a product (BOT). To model P1 and P2, the Pump model was used and HX-F1, HX-F2, and the reboiler (REB) were modeled using the HeatX model. Finally, the DC was simulated by using the RadFrac model. Table 1 shows the detailed specifications of the conventional process model.

### 2.3. Alternative Processes

#### 2.3.1. MEE Process

The MEE process includes several columns, and each reboiler recovers the waste heat of vapor discharged from each column during the evaporation process [20]. Figure 2 shows a simplified diagram of alternative evaporation process model using MEE. First, the MEE process consists of pre-heating and evaporating LA, which is the feed (FEED). During the evaporation process, the MEE columns are arranged such that the pressure is gradually reduced, and the secondary vapor discharged from the previous column is reused as a heat medium for the next column. In the proposed process, during multi-stage distillation, the feed is compressed through a pump and pre-heated through heat exchange of waste water, and is then discharged from the bottom through five distillation columns. The steam used for heat exchange in the first column is combined with the vapor discharged from the top side after the heat exchange process in the flash vessel, and as the temperature of the vapor discharged from the flash vessel is relatively high, it is reused in the heat exchange process. Furthermore, the vapor and waste water discharged from the last flash vessel are refluxed into each distillation column after pre-heating the feed. If the above process is repeated until the last column, then the waste heat of the vapor can be reused remarkably, thereby reducing the amount of steam used in the reboiler as well as the energy consumption of the process. To model P1 and P2, the Pump model was used, and HX-FEED and HX-1–5 wee modeled by employing the HeatX model. Finally, EVA1–5 was simulated by using the RadFrac model. Table 2 shows the detailed specifications of the alternative process model using MEE.

#### 2.3.2. TVR-Assisted Evaporation Process

TVR is a technology for recovering waste heat as useful energy through mechanical compression [21]. Figure 3 provides a simplified diagram of the alternative process model using TVR.

The TVR-assisted evaporation process recovers waste heat from discharged low-pressure vapor by increasing the pressure of the vapor using high-pressure driving steam. By pressurizing the low-pressure vapor and recovering the waste heat with high energy, the consumed steam energy is reduced. In the alternative process model using TVR, a steam ejector (EJECTOR) is used to reduce the amount of steam consumed in the reboiler by recycling the discharged vapor [22]. The water vapor discharged from the tower is mixed with middle-pressure steam (MP-IN), and heat exchange is performed in the column reboiler (REB). The ejector creates a vacuum through rapid pressure change in the motive fluid, sucks in the suction fluid, and then discharges the mixed fluid, which increases the energy of the steam through mechanical vapor compression.

For model P1, the Pump model was used, and HX-F1 and REB were modeled by employing the HeatX model. Then, the DC was simulated by using the RadFrac model. A mixer was used to simulate the steam ejector, and the outlet pressure of the mixer was set so that the pressure of steam discharged through the ejector could be 1.2 kg/cm^2^·g. Table 3 shows the detailed specifications of the alternative process model using TVR.

## 3. Results and Discussion

This section presents the simulation results of the conventional and alternative processes for evaporation in PLA production. To address economic feasibility, we compared the TAC of the alternative and conventional processes.

### 3.1. Simulation Results

When modeling the distillation column in Aspen plus V10.0, the calorific value of the reboiler can be simulated, but not the amount, temperature, or pressure of steam flowing into the reboiler. Therefore, in this study, a virtual reboiler was added to predict the steam usage of the modeled distillation column. By setting the amount of heat calculated in the virtual reboiler and the amount of heat used in the distillation column to be the same, the amount of steam used in the distillation column for each process was predicted. The detailed simulation results are presented in the Appendix A. Table 4 shows the simulation results for the conventional process using SEE.

According to Table 4, approximately 145,440 kg/h of water and 16,160 kg/h of LA in FEED are evaporated in the conventional evaporation process using SEE. Further, 140,056 kg/h of water is evaporated, and the mass flow of water in PRODUCT is approximately 5384 kg/h. The FEED is concentrated to 75 wt% of water. Finally, the calculated steam consumption required to evaporate the FEED is 161,726 kg/h, which indicates significantly high steam consumption. The evaporation process using SEE has an advantage in the low capital cost compared with the evaporation process using MEE or TVR. However, since evaporation process using SEE does not recover the waste heat energy in emitted vapor, the steam consumption to satisfy the water concentration is high. Table 5 shows the simulation results of the alternative evaporation process using MEE.

According to Table 5, approximately 145,440 kg/h of water and 16,160 kg/h of LA in FEED are evaporated in the alternative evaporation process using MEE. Further, 140,101 kg/h of water is evaporated, and the mass flow of water in PRODUCT is approximately 5339 kg/h, resulting in a FEED concentration of 75 wt% in water. The pressures of PRODUCT and VAPOR are 1.2 kg/cm^2^·g and 1.1 kg/cm^2^·g, respectively, indicating lower pressure compared to the conventional process using SEE. This difference exists because in the MEE process, the operating pressure of the column continually decreases from the first column to the last column. In addition, the temperature of VAPOR is 121.83 °C, which is reduced by approximately 14.98% compared to that of VAPOR in the conventional process using SEE. The decrease in temperature of VAPOR indicates that MEE can recover the waste heat of vapor discharged from each column more effectively than SEE. Finally, the calculated steam consumption required to evaporate the FEED is 45,800 kg/h. The evaporation using MEE has high capital cost compared with the evaporation process using SEE because of additional installation of the effects. However, since the evaporation process using MEE can recover the waste heat in emitted vapor though the additional effects, the steam consumption can be decreased by approximately 76.68% compared to that in the conventional process using SEE. Finally, Table 6 shows the simulation results for the alternative evaporation process using TVR.

According to Table 6, 140,053 kg/h of water in FEED is evaporated and the mass flow of water in PRODUCT is approximately 5387 kg/h, resulting in a FEED concentration of 75 wt% in water. The pressures of PRODUCT and VAPOR are −0.25 kg/cm^2^·g and −0.35 kg/cm^2^·g, respectively, which are lower than those in the other processes. In the alternative process using TVR, low-pressure vapor enters with driving steam through the steam ejector. At this time, the speeds of the two fluids increase due to the increasingly narrow shape inside the ejector. Then, very low pressure is formed inside the ejector by the steam injected at high speed and the vapor discharged from the column can be continuously sucked into the steam ejector. In addition, the temperature of VAPOR is 88.82 °C, which is approximately 38.02% lower than that in the conventional process using SEE. The decrease in temperature of VAPOR also indicates that TVR can recover the waste heat of vapor discharged from the column more effectively than SEE. Finally, the calculated steam consumption required to evaporate the FEED is 16,100 kg/h, and the steam consumption can be decreased by approximately 90.04% compared to that in the conventional process using SEE.

### 3.2. Economic Assessment

The objective of a technical economic assessment of a process is to provide a basis for judging the possibility of investment by preliminarily reviewing the economic feasibility at the process development stage [23,24]. In this study, the feasibility of the two proposed processes for PLA evaporation was reviewed to determine the most economical process by comparing the economic feasibility of the two alternative processes with that of the existing process. Then, the TAC of the conventional process and each alternative process was calculated as an economic feasibility evaluation criterion. The TAC is the overall annual cost and is given by
TAC = EAC + AOC(1)
where EAC is the equivalent annual cost and AOC is the annual operating cost [25,26].

#### 3.2.1. Equivalent Annual Cost

The EAC is the annual cost of owning and maintaining an asset determined by dividing the total capital investment (TCI) of the equipment purchase, operations, and maintenance costs by the present value of the annuity factor (AF):(2)$$\mathrm{EAC}=\frac{\mathrm{TCI}}{\mathrm{AF}},$$
where TCI is calculated by summing the fixed capital cost (FCI) and working capital investment (WCI) [27]:TCI = FCI + WCI(3)

FCI refers to the costs, including the equipment purchase, installation, and piping costs, as well as other expenses required for plant construction, and is calculated by adding up the direct costs (C_direct_) and indirect costs (C_indirect_) [28]_:_FCI = *C*_direct_ + *C*_indirect_
(4)

Second, the WCI indicates the capital cost required to maintain stocks of feedstocks, product, etc. The WCI is calculated as 15% of FCI and is given by [28]
WCI = 0.15 × FCI (5)

Then, the AF, used to calculate the present value of a series of future annuities, is defined as follows:(6)$$\mathrm{AF}=\frac{1-\left[\frac{1}{{\left(1+\mathrm{RP}\right)}^{\mathrm{NP}}}\right]}{\mathrm{RP}}$$
where RP is the rate per period and NP is the number of periods. The detailed descriptions of the parameters, equations, and variables related to the EAC calculation are described in the Appendix A. Table 7 shows the calculated EAC of each process.

According to Table 7, the EAC of the conventional process using SEE is approximately USD 1,217,514, and that of the alternative process is approximately USD 3,798,312, representing an increase of 212% compared to the conventional process. This increase exists because an additional column, which is expensive, is required for MEE. Finally, the EAC of the alternative process using TVR slightly increases by approximately 20% in conformity with the installation of the steam ejector.

#### 3.2.2. Annual Operating Cost

AOC is the annual cost of operation of raw materials, electricity, steam, etc. As the mass flows of the feed and product are constant in all processes, the raw material costs are not considered, and the fixed costs, plant overhead costs, and general expenses were also ignored in this study [29]. To calculate the AOC, the steam cost according to the steam consumption of each process estimated through the process model and the electricity cost calculated by the electricity consumption of each process predicted using the Aspen Plus Economic Analyzer are summed:AOC = C_steam_ + C_electricity_(7)

The AOC is calculated by setting the cost of steam at USD 36.36/ton, the cost of electricity at USD 0.065/kW, and the annual operating time at 365 days. Table 8 shows the annual operating time of each process.

The AOC of the conventional process using SEE is approximately USD 51,594,818. In the alternative process using MEE, the AOC is approximately USD 14,839,411 (decrease of 71%) despite the increase in the electricity cost of approximately USD 4247 compared to that in the conventional process. In addition, the AOC of the alternative process using TVR is decreased by approximately 90% due to the reduced steam consumption.

#### 3.2.3. Total Annualized Cost

Finally, the TAC was calculated by summing the calculated EAC and AOC. Figure 4 shows the calculated TCI, EAC, AOC, and TAC.

According to Figure 4, the TCI and EAC of the alternative process using MEE are the highest due to the installation of an additional column, which is expensive, but the TCI and EAC of the alternative process using TVR are slightly increased compared to those of the conventional process using SEE because the steam ejector cost is low. However, the AOC of the conventional process using SEE is the highest because of the large amount of steam consumption. The AOCs of the alternative processes using MEE and TVR can be decreased due to the reduced steam consumption, despite the increased electricity consumption as the waste heat of the vapor discharged from the column can be efficiently recovered. The calculated TAC of the conventional process using SEE is approximately USD 51,598,818, and that of the alternative process using MEE is approximately USD 14,839,411, representing a decrease of 71% compared to the conventional process. Finally, the TAC of the alternative process using TVR is approximately USD 5,199,374, representing a decrease of approximately 90%. Thus, the alternative process using TVR is the cost-optimal process.

## 4. Conclusions

In this work, we designed alternative evaporation processes for PLA production and determined the cost-optimal evaporation process by performing an economic assessment. This study makes two major contributions to the literature. First, to the best of the author’s knowledge, this work represents the first attempt to decrease the cost of evaporation in PLA production by using MEE and TVR to recover waste energy efficiently from emitted vapor. Second, this study will also enable increased PLA usage in many industries, providing environmental protection by reducing the use of plastic, which is produced from petrochemicals, by identifying a cost-optimal alternative evaporation process for PLA production. The alternative MEE and TVR processes reduced the steam consumption by 71.36% and 89.97%, respectively, compared to that of the conventional process. In addition, the economic assessment results show that the cost-optimal process is the alternative process using TVR, which decreased the TAC by approximately 90% compared to that of the conventional process. Therefore, this study provides valuable insight for many PLA production industries, which is anticipated to enable cost-effective increased PLA usage in many industries and environmental protection from petrochemical-based plastics. Further, this paper will interest a broader audience because it addresses key focus areas, such as process design and economic assessment.

## Figures and Tables

**Figure 1 polymers-14-02120-f001:**
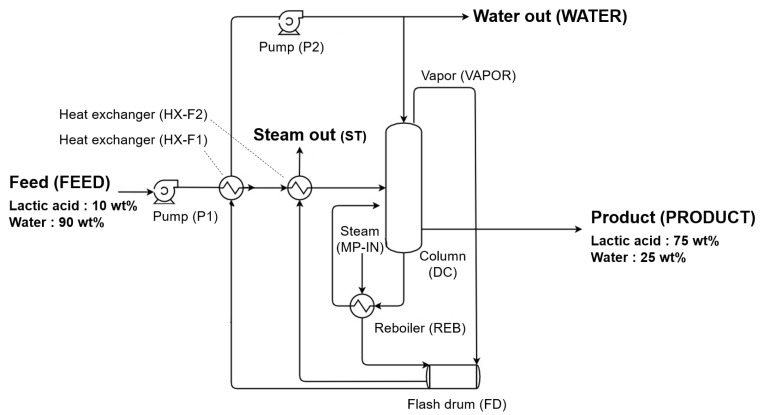
Simplified diagram of conventional evaporation process model using SEE.

**Figure 2 polymers-14-02120-f002:**
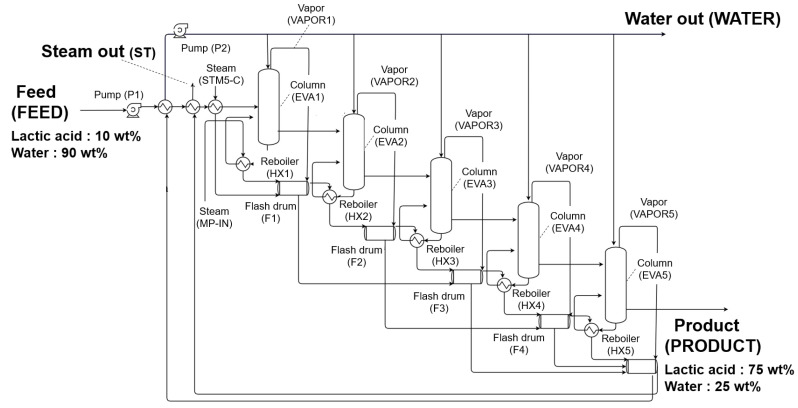
Simplified diagram of alternative evaporation process model using MEE.

**Figure 3 polymers-14-02120-f003:**
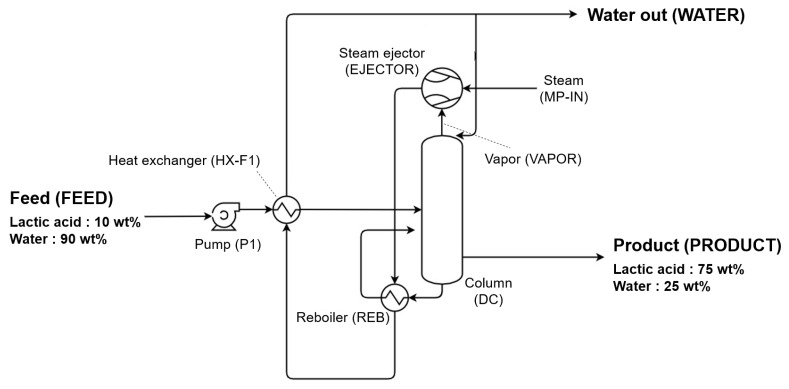
Simplified diagram of alternative process model using TVR.

**Figure 4 polymers-14-02120-f004:**
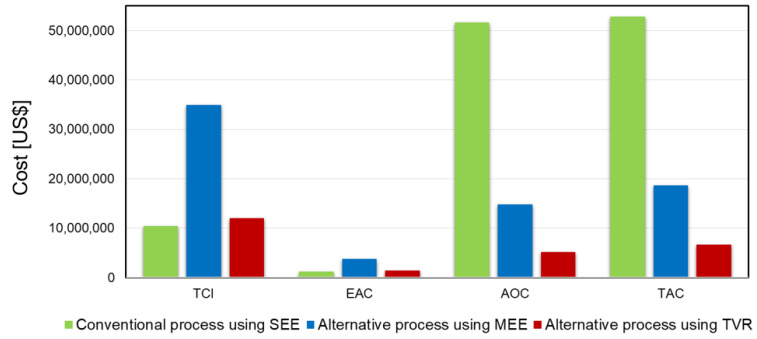
Calculated TCI, EAC, AOC, and TAC of each process.

**Table 1 polymers-14-02120-t001:** Specifications of conventional process model.

Block	Description	Specification	Value	Unit
DC	Distillation column	Distillate rate	147,565	kg/h
		Number of stages	11	
		Stage 1/Condenser pressure	3	kg/cm^2^·g
		Stage pressure drop	0.01	kg/cm^2^·g
		Reboiler outlet molar vapor fraction	0.2	
FD	Flash drum	Pressure	3	kg/cm^2^·g
HX-F1	Heat exchanger	Exchanger duty	10	MMKcal/h
		Minimum temperature approach	5	°C
HX-F2	Heat exchanger	Hot/cold outlet temperature approach	5	K
		Minimum temperature approach	5	°C
P1	Centrifugal pump	Discharge pressure	8	kg/cm^2^·g
P2	Centrifugal pump	Discharge pressure	8	kg/cm^2^·g
REB	Heat exchanger	Hot stream outlet vapor fraction	0.01	
		Minimum temperature approach	5	°C

**Table 2 polymers-14-02120-t002:** Specification of alternative process model using MEE.

Block	Description	Specification	Value	Unit
EVA1	Distillation column	Distillate rate	24,300	kg/h
		Number of stages	11	
		Stage 1/condenser pressure	7.5	kg/cm^2^·g
		Stage pressure drop	0.01	kg/cm^2^·g
		Reboiler outlet molar vapor fraction	0.2	
EVA2	Distillation column	Distillate rate	26,800	kg/h
		Number of stages	11	
		Stage 1/condenser pressure	6.1	kg/cm^2^·g
		Stage pressure drop	0.01	kg/cm^2^·g
		Reboiler outlet molar vapor fraction	0.2	
EVA3	Distillation column	Distillate rate	28,800	kg/h
		Number of stages	11	
		Stage 1/condenser pressure	4.71	kg/cm^2^·g
		Stage pressure drop	0.01	kg/cm^2^·g
		Reboiler outlet molar vapor fraction	0.2	
EVA4	Distillation column	Distillate rate	32,300	kg/h
		Number of stages	11	
		Stage 1/condenser pressure	3.25	kg/cm^2^·g
		Stage pressure drop	0.01	kg/cm^2^·g
		Reboiler outlet molar vapor fraction	0.2	
EVA5	Distillation column	Distillate rate	15,000	kg/h
		Number of stages	11	
		Stage 1/condenser pressure	1.1	kg/cm^2^·g
		Stage pressure drop	0.01	kg/cm^2^·g
		Reboiler outlet molar vapor fraction	0.2	
F1	Flash drum	Pressure	7.5	kg/cm^2^·g
F2	Flash drum	Pressure	6.1	kg/cm^2^·g
F3	Flash drum	Pressure	4.71	kg/cm^2^·g
F4	Flash drum	Pressure	3.25	kg/cm^2^·g
F5	Flash drum	Pressure	1.1	kg/cm^2^·g
HX-1–5	Heat exchanger	Cold stream outlet vapor fraction	0.2	
HX-FEED	Heat exchanger	Exchanger duty	10	MMKcal/h
		Minimum temperature approach	5	°C
HX-STM5, HX-WW	Heat exchanger	Hot/cold outlet temperature approach	5	K
		Minimum temperature approach	5	°C
P1 and P2	Centrifugal pump	Discharge pressure	8	kg/cm^2^·g

**Table 3 polymers-14-02120-t003:** Specifications of alternative process model using TVR.

Block	Description	Specification	Value	Unit
DC	Distillation column	Distillate rate	141,260	kg/h
		Number of stages	11	
		Stage 1/condenser pressure	−0.35	kg/cm^2^·g
		Stage pressure drop	0.01	kg/cm^2^·g
		Reboiler outlet molar vapor fraction	0.2	
HX-F1	Heat exchanger	Hot/cold outlet temperature approach	5	K
		Minimum temperature approach	5	K
P1	Centrifugal pump	Discharge pressure	1	kg/cm^2^·g
REB	Heat exchanger	Hot stream outlet vapor fraction	0.01	
		Minimum temperature approach	5	K
EJECTOR	Steam ejector	Pressure	1.2	kg/cm^2^·g

**Table 4 polymers-14-02120-t004:** Simulation results for conventional process.

Classification	FEED	PRODUCT	VAPOR	MP-IN	Units
Temperature	25	159.19	143.3	179.27	°C
Pressure	0	3.1	3	9	kg/cm^2^_·_g
Mass flow of water	145,440	5384	147,565	161,726	kg/h
Mass flow of LA	16,160	16,151	-	-	kg/h
Total mass flow	161,600	21,535	147,565	161,726	kg/h

**Table 5 polymers-14-02120-t005:** Simulation results for alternative evaporation process using MEE.

Classification	FEED	PRODUCT	VAPOR5	MP-IN	Units
Temperature	25	137.25	121.83	179.27	°C
Pressure	0	1.2	1.1	9	kg/cm^2^·g
Mass flow of water	145,440	5339	35,393	45,800	kg/h
Mass flow of LA	16,160	16,017	-	-	kg/h
Mass flows	161,600	21,356	35,393	45,800	kg/h

**Table 6 polymers-14-02120-t006:** Simulation results for the alternative evaporation process using TVR.

Classification	FEED	PRODUCT	VAPOR	MP-IN	Units
Temperature	25	104.91	88.82	179.27	°C
Pressure	0	−0.25	−0.35	9	kg/cm^2^·g
Mass flow of water	145,440	5387	147,554	16,100	kg/h
Mass flow of LA	16,160	16,160	−	−	kg/h
Mass flows	161,600	21,546	147,554	16,100	kg/h

**Table 7 polymers-14-02120-t007:** EAC of each process.

Classification	Conventional Process Using SEE	Alternative Process Using MEE	Alternative Process Using TVR
**Direct cost on site**			
Purchased equipment	2,064,500	6,897,400	2,383,723
Instruments	970,315	3,241,778	1,120,350
Installation	371,610	1,241,532	429,070
Piping	1,362,570	4,552,284	1,573,257
Electrical	227,095	758,714	262,209
**Direct costs offsite**			
Buildings	371,610	1,241,532	429,070
Yard Improvements	206,450	689,740	238,372
Service facilities	1,445,150	4,828,180	1,668,606
Land	123,870	413,844	143,023
**Indirect cost**			
Engineering	681,285	2,276,142	786,628
Construction	846,445	2,827,934	977,326
Contractor’s fee	141,418	472,471	163,285
Contigency	282,836	944,943	326,570
**Working capital investment**	1,364,273	4,557,974	1,575,224
**FCI**	9,095,154	30,386,495	10,501,494
**TCI**	10,459,427	34,944,470	12,076,718
**EAC**	1,217,514	3,798,312	1,465,557

**Table 8 polymers-14-02120-t008:** AOC of each process.

Classification	Conventional Process Using SEE	Alternative Process Using MEE	Alternative Process Using TVR	Unit
Electricity consumption	146	153	60	kW
Steam consumption	162	46	16	ton/h
Electricity cost	82,888	87,135	34,039	$/year
Steam cost	51,511,930	14,752,276	5,165,335	$/year
**AOC**	51,594,818	14,839,411	5,199,374	$/year

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
