# Peer review of "Design and Economic Assessment of Alternative Evaporation Processes for Poly-Lactic Acid Production"

_polymers, 2022, doi:10.3390/polym14102120_

Round 1
Reviewer 1 Report
The manuscript is interesting work, but before its acceptance needs improvements. Specifically, there are reported only the yields values e any characterization of products. In my view, this is important point for readers, especially, if there are proposed two new methods in comparison to one traditional method for PLA production.
In the conclusion section is written that in this work are discussed two new aspects, but below is discussed only one. This must be corrected/rewritten.
The reference section must be improved and the work aim must be formulated better in the introduction, considering current literature.
Obtained results must be commented more critically
Author Response
Reviewer 1
The manuscript is interesting work, but before its acceptance needs improvements. Specifically, there are reported only the yields values e any characterization of products. In my view, this is important point for readers, especially, if there are proposed two new methods in comparison to one traditional method for PLA production.
1) In the conclusion section is written that in this work are discussed two new aspects, but below is discussed only one. This must be corrected/rewritten.
(Response) Thank you for your detailed comment. We revised conclusion correctly, and the modification is as follows.
On page 18, in section4
This study makes two major contributions to the literature. First, to the best of the author’s knowledge, this work represents the first attempt to decrease the cost of evaporation in PLA production by using MEE and TVR to recover waste energy efficiently from emitted vapor. Second, this study will also enable increased PLA usage in many industries, providing environmental protection by reducing the use of plastic, which is produced from petrochemicals, by identifying a cost-optimal alternative evaporation process for PLA production.
2) The reference section must be improved and the work aim must be formulated better in the introduction, considering current literature.
(Response) We totally agree with your comment. Thank you for your kind comment. We enhanced the reference section to improve the work aim.
3) Obtained results must be commented more critically
(Response) Thank you for your detailed comment. We totally agree with your comment. We enhanced obtained results more critically and the modification is as follows.
On page 12, in section 3.1
Finally, the calculated steam consumption required to evaporate the FEED is 161,726 kg/h which indicates significantly high steam consumption. The evaporation process using SEE has an advantage of the low capital cost compare with evaporation process using MEE or TVR. However, since evaporation process using SEE is not recover the waste heat energy in emitted vapor, thus the steam consumption to satisfy the water concentration is high.
On page 13, in section 3.1
The evaporation using MEE has high capital cost compared with evaporation process using SEE because of additional installation of the effects. However, since the evaporation process using MEE can recover the waste heat in emitted vapor though the additional effects, and thus the steam consumption can be decreased by approximately 76.68% compared to that in the conventional process using SEE.
Sincerely,
Junghwan Kim
Green Materials and Processes Group
Korea Institute of Industrial Technology
55, Jongga-ro, Jung-gu, Ulsan, Korea (44413)
kjh31@kitech.re.kr
(Tel.) +82-52-980-6629
(Fax) +82-52-980-6669

Reviewer 2 Report
Manuscript is very well written, and scientific findings are presented in a coherent manner. Two revisions requested
1) All of the analysis was performed on the unit operation of 'lactide formation' in PLA production. In reality, utilities like steam and electricity are used across multiple unit operations, and often recycled from one unit operation to other. It would be good if the authors add a comment on the complete process as a whole and confirm that the steam is not generated in any other unit operation which might reduce the need of steam in conventional process and influence the current analysis.
2) Page 1, line 37: This sentence is written twice, 'After the reaction has been completed, in the lactide step, the pre-polymer is placed into the lactide reactor for polymerization'
Author Response
Reviewer 2
Manuscript is very well written, and scientific findings are presented in a coherent manner. Two revisions requested
1) All of the analysis was performed on the unit operation of 'lactide formation' in PLA production. In reality, utilities like steam and electricity are used across multiple unit operations, and often recycled from one unit operation to other. It would be good if the authors add a comment on the complete process as a whole and confirm that the steam is not generated in any other unit operation which might reduce the need of steam in conventional process and influence the current analysis.
(Response) We totally agree with your comment. Thank you for your important comment. We add a comment in introduction and the modification is as follows.
On page 3, in section 1
In addition, in many multiple unit operations, utilities such as steam and electricity are used across multiple unit operation and recycled from one unit operation to other but in PLA production, steam and electricity are not generated by other processes, and thus the cost of utilities is significantly high.
2) Page 1, line 37: This sentence is written twice, 'After the reaction has been completed, in the lactide step, the pre-polymer is placed into the lactide reactor for polymerization'
(Response) Thank you for your kind comment. We deleted the sentence correctly.
Sincerely,
Junghwan Kim
Green Materials and Processes Group
Korea Institute of Industrial Technology
55, Jongga-ro, Jung-gu, Ulsan, Korea (44413)
kjh31@kitech.re.kr
(Tel.) +82-52-980-6629
(Fax) +82-52-980-6669

Round 2
Reviewer 1 Report
The paper can be accepted in this form. The v2 is not reported in using Polymers Template, please, use the template.